# Processing Compostable PLA/Organoclay Bionanocomposite Foams by Supercritical CO_2_ Foaming for Sustainable Food Packaging

**DOI:** 10.3390/polym14204394

**Published:** 2022-10-18

**Authors:** Simón Faba, Marina P. Arrieta, Ángel Agüero, Alejandra Torres, Julio Romero, Adrián Rojas, María José Galotto

**Affiliations:** 1Packaging Innovation Center (LABEN), Department of Food Science and Technology, Faculty of Technology, Center for the Development of Nanoscience and Nanotechnology (CEDENNA), University of Santiago de Chile (USACH), Santiago 9170201, Chile; 2Departamento de Ingeniería Química Industrial y del Medio Ambiente, Escuela Técnica Superior de Ingenieros Industriales, Universidad Politécnica de Madrid (ETSII-UPM), Calle José Gutiérrez Abascal 2, 28006 Madrid, Spain; 3Grupo de Investigación: Polímeros, Caracterización y Aplicaciones (POLCA), 28006 Madrid, Spain; 4Instituto de Tecnología de Materiales (ITM), Universidad Politécnica de Valencia (UPV), Plaza Ferrándiz y Carbonell 1, 03801 Alcoy, Spain; 5Laboratory of Membrane Separation Processes (LabProSeM), Department of Chemical Engineering and Bioprocesses, Engineering Faculty, University of Santiago de Chile (USACH), Santiago 9170201, Chile

**Keywords:** foaming, PLA, organoclay, nanocomposites, food packaging

## Abstract

This article proposes a foaming method using supercritical carbon dioxide (scCO_2_) to obtain compostable bionanocomposite foams based on PLA and organoclay (C30B) where this bionanocomposite was fabricated by a previous hot melt extrusion step. Neat PLA films and PLA/C30B films (1, 2, and 3 wt.%) were obtained by using a melt extrusion process followed by a film forming process obtaining films with thicknesses between 500 and 600 μm. Films were further processed into foams in a high-pressure cell with scCO_2_ under constant conditions of pressure (25 MPa) and temperature (130 °C) for 30 min. Bionanocomposite PLA foams evidenced a closed cell and uniform cell structure; however, neat PLA presented a poor cell structure and thick cell walls. The thermal stability was significantly enhanced in the bionanocomposite foam samples by the good dispersion of nanoclays due to scCO_2_, as demonstrated by X-ray diffraction analysis. The bionanocomposite foams showed improved overall mechanical performance due to well-dispersed nanoclays promoting increased interfacial adhesion with the polymeric matrix. The water uptake behavior of bionanocomposite foams showed that they practically did not absorb water during the first week of immersion in water. Finally, PLA foams were disintegrated under standard composting conditions at higher rates than PLA films, showing their sustainable character. Thus, PLA bionanocomposite foams obtained by batch supercritical foaming seem to be a sustainable option to replace non-biodegradable expanded polystyrene, and they represent a promising alternative to be considered in applications such as food packaging and other products.

## 1. Introduction

Environmental concern has increased in the last decade, and consequently, the use of non-renewable petrochemical-based resources should be determined not only by manufacturers but also by consumers. In the same way, packaging materials should not be hazardous for the environment after their useful life. Thus, short-term applications, such as food packaging, should be biodegradable or destined to be reused and/or recycled. Meanwhile, the amount that reaches landfills should be minimized, according to the circular economy model. In this regard, biodegradable and bio-based plastics have been introduced into the field of packaging, hence making it possible to reduce the consumption of non-renewable materials as well as to prevent their accumulation in landfills after their useful life. Altogether they can ensure that nature is preserved for future generations [1,2].

One of the most studied biopolymers for packaging materials is poly (lactic acid) (PLA). PLA is a biodegradable thermoplastic polyester produced from renewable resources, which today is one of the most promising polymers to commercially replace conventional petrochemical-based materials such as LDPE [3]. This biopolymer can be produced from various natural raw materials such as corn or sugar cane, and it can also be degraded under composting conditions by means of a hydrolysis process followed by microorganism attack [2,4,5]. Moreover, PLA can be processed by usual thermoplastic technologies already available in the plastic processing industry (i.e.,: extrusion, injection molding, thermoforming, film forming, and foaming) [6]. In this context, PLA foams have emerged in recent years as a promising alternative for sustainable food packaging and other fast-moving consumer goods compared to conventional polymer foams based on polyolefins such as polystyrene (PS), as they have the already mentioned inherent renewable carbon content and compostability as a sustainable end-of-life option [7,8].

PLA foaming can be carried out by chemical blowing agents, supercritical CO_2_ (scCO_2_) batch processing (i.e., in an autoclave), foam injection molding, or extrusion foaming [9,10,11]. Foaming with supercritical fluids (SCF) is a complex dynamic process that needs a complete understanding of the fundamentals of thermodynamics; physics; the chemistry of solutions, interfaces, and interacting species during the whole process; as well as polymer sciences and process engineering. scCO_2_ is the most commonly used process engineering [12,13,14]. Moreover, scCO_2_ as physical blowing agents is advantageous because it expands rapidly due to a phase change at the blowing temperature and pressure. In general, dense CO_2_ has a high foaming efficiency, is non-toxic, environmentally friendly, non-flammable, and chemically inert, and has high thermal stability [12,15]. Thus, from an industrial point of view, scCO_2_ foaming shows crucial advantages over the chemical foaming processes. Although supercritical CO_2_ foaming process optimization and scale-up are required for the transfer of scCO_2_ technology from laboratory scale level to industrial production of compostable foams, it should be highlighted that scCO_2_ foaming is a simple, flexible and low-cost process, since it allows the use of commercial films to be foamed. Therefore, it is expected that the transfer of scCO_2_ technology for the commercial production of compostable foams can be easily feasible.

On the other hand, in the development of foams it has been observed that the formation of cells depends on the nucleation sites inside the polymer. Some studies have shown that the addition of nanoclays to the polymer matrix provides more heterogeneous cell nucleation sites through local pressure variations created around the nanoparticles, in turn improving the melting strength and storage modulus of the polymer [16,17,18]. Many researchers have demonstrated that the use of nanoparticles both organic (i.e.,: cellulose nanofibers) and inorganic (i.e.,: nanoclays) as nucleation agents can successfully improve PLA thermomechanical and barrier properties. The major challenge for the development of nanocomposites is to achieve a homogeneous dispersion of nanoparticles into the polymeric matrix with the goal to obtain an adequate interfacial adhesion with the matrix [19]. Among nanoparticles, organically modified nanoclays such as Cloisite 30B (C30B) are of particular interest since the dispersion of crystalline layered silicates of montmorillonite (MMT) constituted of stacks of clay platelets with two external layers of silicon oxide tetrahedra with a central sheet of aluminum or magnesium oxide octahedra within the PLA polymeric matrix is relatively homogeneous. This is due to the fact that in organically modified montmorillonites (OMMT), the alkali counter-ions have been exchanged with cationic-organic surfactants that make OMMT more compatible with hydrophobic PLA, leading to a nanocomposite with improved overall performance [20,21]. Therefore, it is expected that the use of nanoclays favors the formation of smaller cells (nanocells) compared to conventional PLA foams. In fact, it was demonstrated that by adding only 1 wt.% of nanoclay (Cloisite 30B) as a nucleation agent into the PLA matrix, the foam density can be reduced, and an increase in the modulus is also observed. In addition, using Cloisite 30B nanoclay as the nucleation agent allows obtaining small and uniform cells [22]. In fact, the cell sizes in PLA foams can be adjusted from micro to nano scale by varying the amount of nanoclay incorporated into the polymeric matrix [23,24]. Consequently, the operating conditions and the characteristics of the fillers, such as their size, shape factor, and surface chemistry, play an important role in the final foam morphology [12], allowing the production of foams with specific desired properties.

Biopolymer foams are commonly obtained by foam processing in batches previously obtaining pellet or films through casting processes [25,26], injection molding [27], compression molding [28,29] or melt extrusion [23,30,31]. However, with the aim to transfer the PLA foaming process from the laboratory to the industrial scale, films made by high-productivity techniques, such as extrusion, should be converted into foams.

In this work, the effect of different nanoclay loading levels (C30B in 1, 2 and 3 wt.%) on the morphological, structural, thermal, and mechanical properties of melt-extruded PLA foams obtained by the supercritical foaming batch process was evaluated with the aim of the proposed process as a simple, quick, and scalable method to obtain cell-closed PLA foams intended for sustainable food packaging applications. Since the moisture gain in food contact materials is a critical issue and water uptake leads to dimensional instability of the packaging materials, water absorption measurements were conducted. Finally, since these foams are intended for the biodegradable food packaging sector, their disintegration under composting conditions was conducted at a laboratory-scale level.

## 2. Materials and Methods

### 2.1. Materials

Poly (lactic acid) (PLA), 2003D (specific gravity 1⁄4 1.24; MFR g/10 min (210 °C, 2.16 kg)), was purchased from Natureworks^®^ Co. (Plymouth, MN, USA); Minnetonka (Minneapolis, MN, USA). The commercial organoclay-modified (montmorillonite), Cloisite^®^ 30B (C30B) (100 meq/100 g) was provided by Southern Clay Products (Texas, TE, USA). Carbon dioxide (CAS No: 124-38-9) was supplied by Linde (Santiago, Chile). 

### 2.2. Preparation of PLA Films and Bionanocomposites 

PLA films without additives and with organoclay C30B (1 wt.%, 2 wt.% and 3 wt.%) were prepared by using a co-rotating twin-screw extruder Scientific Labtech^®^ LTE20 (Samutprakarn, Thailand). The temperature profile of the extruder was from 175 to 200 °C (from die to hopper), at a screw speed of 30 rpm, and the films were then collected in a Scientific Labtech^®^ LBCR-150 chill roll attachment (Samutprakarn, Thailand) at 1.8 rpm. PLA powder as well as the C30B clay were previously dried at 60 °C for 24 h. The resulting films presented thicknesses between 500 and 600 μm measured by a digital micrometer Mitutoyo ID-C112 and were stored in a desiccator until being used for the supercritical foaming process.

### 2.3. Supercritical Foaming of PLA Composites

Supercritical fluid foaming of PLA films and bionanocomposite PLA films loaded with 1 wt.%, 2 wt.% and 3 wt.% of C30B was performed using a system schematically described in Figure 1. This process was carried out in the Laboratory of Membrane Separation Processes (LabProSeM) of the University of Santiago de Chile. A 100 mL high-pressure cell was used and loaded with scCO_2_ by means of a Teledyne^®^ ISCO 260D high-pressure pump and an ISCO 500D syringe pump (Lincoln, NE, USA) operated at a constant pressure rate during the foaming runs. The experiments were carried out under constant conditions of pressure (25 MPa) and at a temperature of around 130 °C inside the cell according to preliminary studies reported by Jeong et al. [32]. The temperature of the high-pressure cell was controlled using a thermostatic electric resistance around the cell. The temperature inside the high-pressure cell was controlled by an electro-thermal resistance strip around the cell. The process to obtain the foam from the film once the pressurized (25 MPa) and temperature conditions (130 °C) were reached inside the cell took 30 min. Subsequently, once the saturation of the PLA films with CO_2_sc was reached, the high-pressure cell was depressurized to atmospheric pressure by quickly releasing CO_2_ within 1 s and the system was stabilized through convective air chilling at room temperature (25 °C approximately).

### 2.4. Film and Foam Characterization 

#### 2.4.1. Viscosity Molecular Weight

Firstly, the intrinsic viscosity [η] of all the PLA bionanocomposite films and foams was determined by the measurement of its capillary viscosity using a Ubbelohde viscometer (type C) according to ISO 1628. All the samples were diluted in chloroform (Sigma-Aldrich 99% purity) and measured keeping the viscometer in a water bath with a controlled temperature of 25 °C and at least five concentrations of each sample were used. Then, the viscosity molecular weight (Mv) can be estimated by means of the Mark–Houvink Equation (1):(1)$$\left[\eta \right]=K\times M_{v}^{a}\;$$
where *K* and a are 1.53 × 10^−2^ and 0.759 respectively, for PLA [33].

#### 2.4.2. Morphological Analysis

Film samples, previously frozen in liquid N_2_, were cryofractured and sputtered with a gold palladium alloy layer to be observed by FESEM in a Supra 55-Zeiss.

Cell structures were analyzed by Scanning Electron Microscopy (SEM), using a JSM-5410 Jeol Scanning Microscope with accelerating voltage at 10 kV. Samples were previously frozen in liquid nitrogen, cryo-fractured and then coated with gold/palladium layer using a Sputtering System Hummer 6.2 cathode spraying system, and SEM micrographs of surface and cross-section samples were taken.

In order to obtain the cell size distribution, the size (d) and mean cell wall thickness (δ) of at least 100 cells in the core part of the cross-section of the fractured foam was measured by digital image analysis, using ImageJ software. The cell densities (NC) were calculated by Equation (2) [23]. The mass density of both pre-foamed ($\rho_{p}$) and post-foamed ($\rho_{f}$) samples in g cm^−3^ were estimated by using the method of buoyancy by submerging sample in a graduated cylinder of water and observing the increase in volume, following Archimedes’ principle [23].
(2)$$NC=\frac{1-\rho_{f}/\rho_{p}}{10^{-4}\;\cdot \;d^{3}}$$

Meanwhile, the porosity or void fraction ($V_{f}$) was given by Equation (3) [29].
(3)$$V_{f}=1-\rho_{f}/\rho_{p}$$

#### 2.4.3. Fourier Transform Infrared (FTIR)—Attenuated Total Reflectance (ATR) Spectroscopy

FTIR–ATR spectra of developed foams as well as neat and bionanocomposites PLA were recorded using a Bruker Alpha spectrometer (Wismar, Germany) equipped with an attenuated total reflection diamond crystal accessory (Bruker, Platinium). This analysis was used to characterize the presence of specific chemical groups in the developed materials. The spectra were obtained with a resolution of 4 cm^−1^ in a range of wave numbers from 4000 to 400 cm^−1^ with 100 scans. Spectrum analysis was performed using OPUS software version 7 (Bruker, Ettlinger, Karlsruche, Germany). To obtain comparable results, spectra were normalized using the absorbance value at 1455 cm^−1^ assigned to the asymmetric bending of CH_3_ group and known to be a suitable as internal standard for PLA [34].

#### 2.4.4. X-ray Diffraction (XRD)

Crystalline phases of PLA bionanocomposite films and foams were studied by means of X-ray diffraction using Siemens D5000 diffraction equipment. The samples were mounted on an appropriate sample holder and surface scanning was performed. The patterns for profile fitting were obtained using CuKα radiation at the 2θ° scanning angle, between 2° and 30°, with a scanning step of 0.02°, at a collection time of 10 s per step.

#### 2.4.5. Thermal Properties 

Differential Scanning Calorimetry (DSC) analysis was performed using a Mettler Toledo DSC 822e International colorimeter (Schwerzenbach, Switzerland). Thermal analysis experiments were evaluated to analyze the foaming process in relation to the modification of the polymeric crystalline structure. Normally, aluminum capsules were prepared containing between 5.0 and 8.0 mg of samples from each run; however, in this study, due to the low density of foams, they were prepared between 1.0 and 2.0 mg. Samples were heated from 25 to 200 °C at a heating rate of 10 °C min^−1^ under a nitrogen atmosphere to avoid hermos-oxidative degradation of the samples. 

Thermogravimetric analysis (TGA) tests were performed using a Mettler Toledo Gas Controller GC20 Stare System TGA/DCS (Schwarzenbach, Switzerland). Samples were heated from 30 to 600 °C at 10 °C min^−1^ under a nitrogen atmosphere with a flow rate of 50 mL min^−1^ to avoid hermos-oxidative degradation of the PLA samples. Decomposition temperature was evaluated at 5% of mass loss (Td, 5%) and the maximum degradation temperature (T_max_) was determined from the first derivative curve. TGA analysis allowed to determine the type of degradation and to evaluate the thermal stability of the different PLA foams.

#### 2.4.6. Mechanical Properties

Mechanical properties were measured with a Shimadzu AGS-X 100N tensile testing machine (Shimadzu Corporation, Kyoto, Japan) equipped with a 100N load cell, at a crosshead speed of 10 mm min^−1^ and initial length of 30 mm. Experiments were carried out on rectangular samples with a size of 5 × 50 mm and at least five specimens were tested for each formulation. The average percentage elongation at break (ε%), Young modulus € and tensile strength (TS) were calculated from the resulting stress–strain curves. 

#### 2.4.7. Water Absorption Analysis

Water absorption of PLA bionanocomposite films and foams here developed was analyzed as recommended by the standard ISO 62:2008. For this, square samples of 15 × 15 mm^2^ were immersed in deionized water for a period of 81 days in a room with a controlled temperature of 23 ± 1 °C. During the test period, once a week samples were extracted from the bath, dried with an absorbent cloth, and weighed on an analytical balance QUINTIX125D-1S analytical balance (Sartorius, Gotinga, Germany) with an accuracy of 0.00001 g. The samples then were again immersed in the deionized water. To ensure the accuracy, tests were carried out in triplicate. The percentage of absorbed water ($\Delta m_{t}$) was calculated following Equation (4):(4)$$\Delta m_{t}\left(\%\right)=\left[\frac{W_{t}-W_{0}}{W_{0}}\;\right]\times 100$$
where $W_{t}\;$ is the sample weight after an immersion time t and $W_{0}\;$ is the initial weight of the dry sample before immersion.

#### 2.4.8. Disintegration under Standard Composting Conditions

Disintegration under composting conditions was performed by following the ISO-20200 standard [35]. Solid synthetic waste was prepared by mixing 10% of compost commercial, 30% rabbit food, 10% starch, 5% sugar, 1% urea, 4% corn oil, and 40% sawdust and it was mixed with water in 45:55 ratio. Water was added periodically to the reaction container to maintain the relative humidity in the compost [36]. PLA film and bionanocomposite samples were prepared in squares of 1.5 × 1.5 cm and they were buried 6 cm in depth in plastic reactors containing the solid synthetic wet waste. Each sample was contained in a textile mesh to allow its easy removal after treatment, but allowing the access of microorganisms and moisture at the same time as it allows to recover the disintegrated samples [37]. Reactors were introduced in an air circulation oven (Memmert GmbH, Schwabach, Germany) at 58 °C for 30 days. The aerobic conditions were guaranteed by periodical gentle mixing of the solid synthetic wet waste [21,36,37]. Films were recovered from the disintegration container at different times (1, 3, 7 and 14 days), washed with distilled water, dried in a vacuum-oven at 40 °C for 24 h, and re-weighed in a QUINTIX125D-1S analytical balance (Sartorius, Gotinga, Germany). Photographs were taken of all samples once they were extracted from the composting medium to qualitatively check the visual aspect of the samples during the disintegration phenomenon [37]. Meanwhile, the disintegration degree was quantified by normalizing the sample weight at each day of incubation to the initial weight and 90% of disintegration was considered as the goal of samples’ disintegrability [33,38,39].

### 2.5. Statistical Analysis

Data analysis was carried out using Statgraphics Plus 5.1 (StatPoint Inc., Herndon, VA, USA). This software was used to implement variance analysis with the ANOVA test. The experimental design was random-type. Statistically significant differences were considered significant for *p* < 0.05, with a 95% confidence level.

## 3. Results

### 3.1. Processing of PLA Bionanocomposite Films and Foams

PLA film bionanocomposites with different C30B contents (1.0 wt.%, 2.0 wt.% and 3.0 wt.%) were prepared by melt-blending by means of an extrusion process directly followed by a film-forming process, as was described in Section 2.2. Figure 2a shows samples of film containing C30B. As shown in Figure 2a, homogeneous and transparent films were obtained, indicating that no visible agglomerations of C30B can be observed in the samples produced by the extrusion process. PLA bionanocomposite films were subjected to scCO_2_-assisted foaming (described in Section 2.3). The temperature was assessed in ranges from 125 °C to 135 °C and the system pressure in ranges from 10 to 25 MPa. Preliminary tests were able to obtain a saturation rate of scCO_2_ at 30 min under conditions of 130 °C and 25 MPa. The depressurization rate was kept constant at 1 second. Figure 2b shows the images of the foams obtained from neat PLA and PLA bionanocomposites. As can be seen, the resulting foams were white due to the rearrangement of the polymer chains and its effect on the refraction of light. Furthermore, asymmetric shapes were observed due to biaxial growth during the foam expansion.

### 3.2. Viscosity Molecular Weight

Table 1 gathers the results obtained for the intrinsic viscosity and the estimated molecular weight calculated by the Mark—Houwink Equation (1). The temperature and shear to which the material is subjected during the processing may cause certain thermal degradation and a diminution of the M_v_ value [33]. In this sense, respecting the M_v_ value of the PLA pellet, which was ~186,000 g mol^−1^, for the PLA-processed film developed in this work a reduction of around 10% was observed; whereas, the addition of the C30B particles further increases this M_v_ reduction with increasing content of C30B. Particularly, in the case of PLA 3% C30B, the highest reduction of 33% with respect to the PLA pellet and a reduction of 15% in regard to PLA film were observed. This effect on the molecular weight of the PLA matrix when C30B are used as reinforcement particles have been reported previously in other works [40]. In this regard, Iturrondobeitia et al. ascribed this behavior to the rapid water absorption characteristic of the nanoclays, which induce the hydrolysis phenomenon that occurs as well in the composite processing. Similarly, Zhou and Xanthos (2009) also observed a reduction in the molecular weight of C30B-loaded PLA and ascribed these changes to the thermal and hydrolytic degradation effects as well as to the extent of the filler dispersion, which they also related to the presence of organomodifiers on the filler surface that may act as catalysts leading to higher reduction of the molecular weight during processing [41].

Although the foaming process by supercritical fluid also reduced somewhat the molecular weight, this reduction was less marked than the extrusion process. The CO_2_ diffusion inside the polymer matrix which produces the formation and growth of the cell together with the temperature and pressure used for the foam production are factors which can modify the viscosity and molecular weight of the PLA. Taking into account that the foaming process was performed after the film extrusion, the final M_v_ value for the neat PLA foam (PLAf) decreased to 144,750 g mol^−1^, which is around 14% less than neat PLA film. For the PLA-C30B bionanocomposite foams, the presence of an external nucleating agent (e.g., solid particles such as nanoclays, nanocrystals, etc.) provides sites with lower surface energy at the polymer–particle interface and promotes higher cell formation rates [42]. Therefore, increasing the cell nucleation during the foaming process, the CO_2_ molecules can diffuse easily into the polymer matrix and are more homogeneously distributed; hence, the modification over the molecular weight is less marked than in the case of neat PLA foam. A noticeable result can be observed for the PLAf 2% C30B, whose M_v_ value is the lowest, although being of the same magnitude order. Nevertheless, it should be mentioned that the morphological and foam parameters discussed previously show that the PLA foam with 2% C30B content was the more heterogeneous structure obtained. Thus, it could be proposed that the biggest mean cell size and the lowest cell density sites where CO_2_ interacts with the material are less but more profitable, and this is reflected in a higher reduction of the molecular weight.

### 3.3. Morphological Results

The incorporation of the nanoclay and its distribution in the polymer matrix structure of the different formulations can be evaluated by means of SEM analysis. SEM micrographs of the cross-section of freeze-fractured samples of all film formulations are shown in Figure 3. PLA films showed a mostly homogeneous surface with no apparent phase separation; the SEM micrographs of the cryofractured surface of the bionanocomposites (Figure 3b–d) are rougher than those of the unfilled PLA films (Figure 3a); there is a different texture due to the presence of C30B, which is more prominent in the sample reinforced at 3 wt.%. The increasing roughness due to the higher contents of C30B is in good agreement with already reported C30B reinforced PLA films [21].

The effect of organoclay addition on the cell morphology of the PLA foam and PLA bionanocomposite foams has been already studied in a previous work by Rojas et al., 2022 [43] and it is shown in Figure 4, while in this work we have studied the cell diameter distribution based on a Gaussian distribution approximation. SEM microscopies of the cross-section and their corresponding cell distribution diameters of PLA foams are depicted for neat PLAf (Figure 4A), PLAf 1% C30B (Figure 4B), PLAf 2% C30B (Figure 4C), and PLAf 3% C30B (Figure 4D). The cell diameter distribution was determined based on a Gaussian distribution approximation. PLA foams evidenced a closed cell structure and presented an average diameter between 21.6 μm and 26.5 μm; this could be due to the different levels of scCO_2_ saturation inside the polymer matrix and C30B amounts. From micrographs it can be concluded that the cell structure was influenced by the presence of nanoclays, obtaining in general terms more uniform, smaller, and well- distributed cells in the PLA-C30B foams. This may be due to the presence of nanoclays which induced the crystal formation, acting as an effective nucleating agent and increasing the PLA’s low melt strength [12]. Several studies have reported that these nanoparticles can act as heterogeneous nucleating agents allowing the formation of cells with closed structures (in most cases), besides acting on their stabilization during biaxial stretching in cell growth (foam expansion), due to their alignment along the cell walls, and increasing their resistance to the stretching force, thereby inhibiting cell rupture. In addition, the cell sizes in PLA foams can be adjusted from micro to nano cell sizes by varying the amount of nanoclay incorporated into the polymeric matrix [23,44]. Compared with the SEM micrographs, the cell size of the PLA foam changed when the organoclay nanoparticles were incorporated. It could be seen that the sample of neat PLA foam has a smooth fractured surface compared to the foam of PLA bionanocomposites. The neat PLA foam exhibited a uniform cell structure and thin cell walls (Figure 4A), in contrast to bionanocomposite PLA foams (Figure 5B–D). The reduction in cell size of bionanocomposites and the increase in cell density are considered to be the result of the cell nucleation effect of C30B nanoparticles during the foaming process of PLA foam bionanocomposites. During the formation of the PLA foams, the organoclay nanoparticles act as nucleating agents able to produce more cell formation sites and, thus, PLA bionanocomposite foams produce smaller cells than neat PLA foam [32]. Furthermore, in Table 2 an organoclay addition effect was observed when the C30B content was higher than 2 wt.%. The cell density increased from 4.9 × 10^11^ cell/ cm^3^ to 8.2 × 10^11^ cell/cm^3^. With increasing C30B content, cell size decreased to 21.6 μm and wall thickness increased to 486 nm. A decrease in porosity ($V_{f}$) was observed in the PLAf foam 3% C30B ($V_{f}$ = 0.83), compared to the neat PLA foams which reaches 90% ($V_{f}$ = 0.90). Therefore, the space inside a foam cell is larger in neat PLA foams than in the case of bionanocomposite PLA foams. Moreover, all the foams obtained were classified as microfoams because their cell sizes resulted in average diameters between 1–100 um, in agreement with those obtained by previous researchers [45,46].

In conclusion, the formation of PLA composites with C30B nanoclay resulted in foams with a more homogeneous cell size and higher cell density compared to virgin PLA foams. The results indicated that the foaming parameters, including the foaming pressure, temperature, and scCO_2_ dissolution time, influence the morphology [29]. It was found that adding 1 wt.% of C30B nanoclay resulted in a foam with fine cell size, but the lowest cell density among the samples and with closed-cell structure.

Right side: cell diameter distribution of PLA foams: (A’) neat PLAf, (B’) PLAf 1% C30B, (C’) PLAf 2% C30B, and (D’) PLAf 3% C30B nanocomposites.

Right side: cell diameter distribution of PLA foams: (A’) neat PLAf, (B’) PLAf 1%C30B, (C’) PLAf 2%C30B and (D’) PLAf 3% C30B nanocomposites.

### 3.4. FTIR Spectra Results

Figure 6 shows the FTIR spectra of neat PLA and PLA-based nanocomposite foams containing organoclay C30B. PLA samples presented their characteristic peaks, such as the carbonyl vibration at approximately 1747 cm^−1^, and peaks at 1040 cm^−1^, 1080 cm^−1^, and 1180 cm^−1^, associated with C–O stretching, C = O, and C–O symmetric stretching and C–O–C stretching, respectively [46]. The carbonyl group, centered at 1753 cm^−1^ in the neat PLA sample, was shifted to somewhat lower wavenumbers in bionanocomposites (1748 cm^−1^) probably due to the crystalline carbonyl stretching as a consequence of the C30B nucleating effect. Due to the fact that nanoclays were incorporated at low concentrations, all PLA nanocomposites presented similar spectra corresponding to PLA characteristic bands. Additionally, the vibration bands corresponding to the stretching of hydroxyl groups and cations from the octahedral sheet in the region of 3100 and 3500 cm^−1^ were not observed (not shown), and the small bands at 2955 and 3004 cm^−1^ are a result of the presence of water adsorbed on the polymer surface [47]. This can be corroborated by FTIR analysis revealing that the carbonyl stretching intensity band decreases with the presence of Cloisite (Figure 7), which indicates that the oxygen of carbonyl group is involved in hydrogen bonding interactions with the hydroxyl groups of the organic modifier of C30B [21].

In conclusion, the samples of neat PLA and PLA-based bionanocomposites did not present significant differences in their chemical structure, which suggests that the processing of batch foaming by scCO_2_ did not modify its chemical structure to any great extent.

### 3.5. X-ray Diffraction (XRD)

X-ray diffraction patterns were used to study the crystalline structure of C30B loaded films and foams. It is well known that the when the silicate layers are uniformly and completely dispersed in a continuous polymeric matrix, an exfoliated or delaminated structure is achieved. The intercalation of the polymeric chains typically increases the interlayer spacing of the organoclay leading to a shift of the diffraction peak towards lower angle values [48]. Figure 8 shows the X-ray diffraction pattern of Cloisite 30B, PLA films as well as PLA-based foams and their nanocomposites. As expected, all transparent film samples showed a mainly amorphous structure with an amorphous halo centered at 2θ = 16.5°, characteristic of the reflection of PLA crystals corresponding to the (203) lattice plane [49]. OMMT powder shows the typical peak of C30B centered at 2θ = 4.82° [50] corresponding to d-spacing (d_001_) to a mean interlayer space of 1.86 nm [51] (see diffractogram a in Figure 8), while the C30B-loaded PLA films display a peak at 2θ = 2.62° (see diffractograms b, c and d in Figure 8) and in the nanocomposites leads to an increase in the d_001_ basal spacing [52]. This shift to lower 2θ angle has been ascribed to the strong interaction between carboxyl groups of the PLA and hydroxyl groups of the organic modifier of C30B as was already mentioned in the FTIR analysis, suggesting that some PLA chains were able to be inserted inside the galleries of the OMMT, increasing the basal distance between the layers with a formation of bionanocomposites characterized by intercalated structures as was demonstrated in a previous work [21]. Moreover, the intensity of the peak at 2θ = 4.82° increased with increasing content of C30B in the formulation. There is another peak centered at around 2θ = 5.3°, ascribed to a clay gallery collapsing and/or the d_002_ reflection [52], that appears as a shoulder in PLA 1% C30B bionanocomposite film and becomes a peak with a higher intensity in bionanocomposites with higher content of C30B (PLA 2% C30B and PLA 3% C30B), which indicates there are some OMMT that are partially exfoliated. 

In the case of the bionanocomposite foams, the amorphous halo disappeared, showing a clear more crystalline diffraction pattern (see diffractograms f, g, h, and i in Figure 8). There are two peaks centered at 2θ = 16.6° and at 2θ = 18.3°, characteristic of the crystalline forms of PLA [53], which correspond to the (203) lattice plane [51]. Moreover, the bionanocomposites processed into foams showed a higher shift of the peak centered at 2θ = 4.82° in pristine C30B to even lower 2θ angles than in the case of bionanocomposite films and also with a higher intensity, showing a greater effect of nanoclay on the bionanocomposite structures when they are processed into foams. The large displacements of these peaks has been already observed by Lee and Hanna (2009), who developed C30B-loaded starch/PLA bionanocomposite foams and ascribed this behavior as indicative of intercalation of a small amount of polymeric chains into the gallery spacing between the silicate platelets [54]. In the present work, the peaks that appeared at around 2θ = 5.3° in bionancomposite films mainly disappeared in bionanocomposite foams, suggesting that the exfoliation was achieved. It seems that scCO_2_ produced a plasticizing effect into the bionanocomposites matrices, increasing the polymer chains’ mobility, improving the dispersion of C30B into the polymeric matrix, and allowing higher interaction between PLA and C30B. Additionally, there is a small peak at around 2θ = 29°, which had been related to the formation of α-form PLA crystals and related to the processing conditions [55], that can be ascribed to the plasticization effect of CO_2_ into the PLA matrix, promoting a nucleation effect.

### 3.6. Thermal Properties

DSC results obtained during the first heating process are shown in Figure 9. The neat PLA film samples showed the glass transition temperature around 60 °C, the cold crystallization temperature at ~116 °C, and the melting peak at around 150 °C. Additionally, a small but clear exotherm peak took place just before melting endotherm ascribed to the increased polymer chains’ mobility, in which the polymer crystallizes and melts upon heating. This behavior has been ascribed to the transition of the disorder (α’) to order (α) PLA crystals, and it is indicative that a large part of the polymeric matrix is in an amorphous state [37]. In fact, from the visual appearance of the films it can be seen that the PLA and PLA-C30B-based bionanocomposite films were completely transparent (Figure 2). 

A slight decrease in the glass transition temperature and the cold crystallization temperature of PLA film nanocomposites was observed. The small changes occurring in the Tg value in bionanocomposite based on PLA with C30B nanoclay are associated with the peak of the enthalpy relaxation of the amorphous zones of the polymer [56]. A somewhat larger shift to lower values of the cold crystallization temperature (Tcc) was observed, which could be due to a nucleation effect of the C30B crystals in the PLA matrix. 

Regarding the foams, in all samples the cold crystallization temperature did not take place, suggesting that the material completely crystallized during the supercritical CO_2_ process. Only in the PLA 3% C30B sample it was observed a significant increase in Tm with respect to the neat PLA foam. It is worth noting that the exotherm area immediately prior to the melting process disappeared with the addition of the C30B nanoparticles. On the other side, PLA 1% C30B and PLA 3% C30B bionanocomposite films melted, showing two peaks suggesting the presence of lamellar populations of crystals with two different crystalline phases. Whereas, in PLA 2% C30B bionanocomposite film there was only one melting peak, suggesting that C30B is well-dispersed into the PLA matrix, able to produce a homogeneous polymeric material. Meanwhile, the neat PLA foam showed only the melting peak around 150 °C, suggesting that PLA completely crystallized during the foaming process. A very similar pattern was shown by the PLA 1% C30B sample, while the PLA 2% C30B and PLA 3% C30B bionanocomposites showed two clearly defined melting peaks, one centered at around 150 °C and the other one centered at around 160 °C, that can be ascribed to the as-formed PLA crystals during film forming process and re-crystallized PLA crystals formed during the foaming process, respectively.

TGA curves and TGA parameters are shown in Figure 10 and Table 3, respectively. TGA parameters showed the degradation temperature (Td, 5%) at around 334 °C and the maximum degradation temperatures (T_max_) around 365 °C; these result is in good accordance with those previously reported in the bibliography [21,46]. In general, PLA-C30B-based bionanocomposites mainly maintained the Td, 5%, while the Tmax was slightly shifted to higher values in bionanocomposites. Different behavior was observed for the thermal parameters of all PLA-based bionanocomposite foams, which showed a clear improvement of the thermal stability of the bionanocomposite with respect to neat PLA. In fact, the Td, 5% degradation temperature of PLA-C30B-based bionanocomposite foams were considerably higher than that of neat PLA foam, around 25 °C, due to the good dispersion of C30B nanoclay into the bionanocomposite foams as a consequence of the CO_2_ presence, which plasticized the system and improved the dispersion. 

### 3.7. Mechanical Properties 

Food packaging materials are required to maintain their integrity to resist the stress that occurs during transport, handling, and storage. Thus, the determination of mechanical properties is of fundamental importance in materials intended for food packaging. In the present work, the mechanical behavior of bionanocomposite PLA foams as a function of C30B loading was evaluated and tensile test results are reported in Figure 11. The production of foamed material produced a more flexible material than the film counterpart. In this sense, the elongation at break of PLA foam increased with respect to the neat PLA film, which showed an elongation at break of 12 ± 2% [21], due to the already mentioned plasticization effect of CO_2_ during the supercritical foaming process. Meanwhile, the Young modulus and tensile strength were reduced with respect to the PLA film as a result of foaming because of the less solid matrix available to support the load and as frequently occurred in plasticized materials [57]. The PLA foam showed comparable values to those observed for PS foams [58]. The C30B-loaded PLA foams clearly showed increased elongation at break, but also increased Young modulus as well as tensile strength with respect to neat PLA foam. This result is due to the good dispersion of the nanoclays into the PLA polymeric matrix achieved during the foaming process, in which the well dispersed nanoclays are exfoliated within the polymeric matrix and are able to enhance the interfacial adhesion of the bionancomposites. Thus, bionanocomposite foams showed improved overall mechanical properties.

### 3.8. Water Absorption Analysis

This study proposed materials based on PLA and C30B nanoclays as a promising application for the food-packaging sector. Among other factors, the material behavior against the humidity and/or contact with water is a key factor for materials intended to be in contact with food. Hence, the analysis of their water absorption capacity and a comparison with the counterpart films was necessary. The evolution of the water uptake of the PLA and bionanocomposite PLA films and foams was evaluated during an immersion time of 84 days at a controlled temperature of 23 ± 1 °C. Firstly, in Figure 12 are plotted the curves of weight gain (wt. %) due to the water absorption versus the immersion time of the PLA film and the PLA-C30B bionanocomposites films. Graphically one can observe that during the first two days, all the films showed a significant increase in mass. Then the weight increment by water absorption continues slower as is common in polymer-based materials [59]. The lowest values are obtained for the neat PLA film in which after three months the mass remained increased by ~1.2%. Although PLA-C30B bionanocomposites’ curves apparently do not become completely asymptotic, the water uptake seems to trend toward saturation mass, which is not reached after 84 days of immersion. However, after day 70 all the mass increment curves are very close to its saturation value. The incorporation of the C30B particles in the PLA film induces a more hydrophilic behavior, showing all the bionanocomposites higher values compared to neat PLA, approaching a 2% mass increment. This is in total agreement with the micrograph discussed above where a rough surface and the apparition of some cavities spread throughout were observed for the PLA-C30B samples.

The supercritical fluid foaming to which PLA and PLA-C30B bionanocomposites were subjected has a clear effect on the hydrophilicity. The water absorption curves of the PLA and bionanocomposite foams are plotted in Figure 13. In a single glance one can see that values reached are extremely high compared to the film samples. The formation of a porous 3D-structure by the foaming process supposes an easy path for the water molecules’ diffusion. This is evidenced in the weight gained by water absorption, which is increased for neat PLA to approximately 150%. Regarding the presence of C30B nanoclay in the foams, a small content of 1 wt.% (PLAf 1% C30B) does not appear to have significant effects on performance against the immersion in water with respect to neat PLA foam. However, when the content of nanoclay was increased, the saturation mass rapidly grew to 300% and 425% for the 2% and 3% of C30B, respectively. This is obviously in total concordance with the increment in the cell density due to the mentioned nucleation effect of the C30B particles in the foam’s production. That is, the clay content increases the hydrophobicity of the material while the foaming process increases the contact area between the water and the material wall (through the pores/cavities). Therefore, the weight gain due to water absorption of the foams depends on the amount of C30B added, as well as on the distribution and size of the cells obtained.

### 3.9. Disintegration under Composting Conditions

PLA film, PLA foam, and their C30B loaded bionanocomposites were disintegrated under standard composting conditions at laboratory scale and their visual appearance during the disintegrability test are shown in Figure 14A, while the mass lost during the disintegrability test as a function of time is shown in Figure 14B.

PLA degradation under composting conditions takes place in two main and consecutive steps: (i) the hydrolytic and (ii) enzymatic degradation processes. From the results, it can be seen that the disintegration phenomenon starts firstly in films. The disintegration of the polymeric matrix under composting conditions starts in the polymeric amorphous phase [60] and this is why films started the disintegration phenomenon previous than the foams. In fact, although the mass loss of films started at around 11 days, the films showed clear signs of disintegration early. In this sense, in only one day the films changed their color and became white, ascribed to the changes in the polymer matrices refraction index due to water absorption and/or presence of products formed during the hydrolytic degradation process [39]. Then, at 7 days the materials became breakable, while the mass loss started at 11 days and at 18 days became opaquer with a clear brown tonality. The foams started the weight loss at 18 days of disintegration due to their more crystalline nature, as was already mentioned in the DSC analysis (Figure 9). In fact, from the water absorption assay it is possible to see that the foams started the water absorption between 10 and 14 days (Figure 13); once the water enters into the structure of the foams, it is able to start the hydrolysis process required for their disintegration in shorter polymeric chains. Nevertheless, once the disintegration phenomenon started rapidly, it led to a higher disintegration rate than in the films. This behavior can be ascribed to the foam structure that possesses higher exposed surface in contact with water and further hydrolyzes faster the polymeric chains into smaller ones to be then enzymatically degraded by microorganisms present in the compost medium. The neat PLA film disintegrated faster than its corresponding bionanocomposites due to the reinforcing effect provided by the C30B nanoclays that delayed the composting disintegration of the PLA matrix, as frequently occurs in biodegradable polymers reinforced with nanoclays [60]. A similar tendency was observed for PLA- and PLA-C30B-based foams, but the PLAf 2% C30B formulation showed a somewhat slower disintegration rate, which is directly related with the high ordered crystals formed during the foaming process as was mentioned in the DSC analysis (as seen in Figure 9B).

## 4. Conclusions

In this work, the foam processing conditions of C30B-loaded PLA-based composites by means of CO_2_ supercritical technology were studied. It was possible to successfully obtain PLA bionanocomposite foams by a simple method of foaming using supercritical CO_2_ from films prepared by a melt-blending approach. The effect of the nanoclays’ incorporation on the morphological, structural, thermal, mechanical, and water absorption properties of the bionanocomposite foams as well as on their disintegrability under standard composting conditions was studied in order to understand the interaction of the nanoclay with the polymer and scCO_2_ during the fabrication process. Thus, the main novelty of this work is to show easy scalability in the fabrication of biodegradable foams because of the industrial nature of the used processing technologies.

This work was focused on the achievement of an optimum combination of industrial scalable processing conditions to increase cell density to maximize the mechanical performance of the obtained end bionanocomposite foams. It was observed that in the supercritical foaming process, the reduction in the molecular weight of the samples was less evident than the observed one in the extrusion process.

On the other hand, through SEM analysis, it was observed that PLA foams loaded with the organomodified clay nanoparticles caused nucleation sites which produced smaller cells than the observed ones in the neat PLA foam. The selected foaming processing conditions have a significant effect on the degree of dispersion and exfoliation of the nanoclays into the polymeric matrix. During the foaming process, heterogeneous nucleation played an important role when the organoclay content was higher than 2 wt.%. The cell density increased from 4.9 × 10^11^ cell/cm^3^ to 8.2 × 10^11^ cell/cm^3^. With increasing C30B content, cell size decreased to 21.6 μm and wall thickness increased to 486 nm. Measurements of the FTIR spectra showed that the incorporation of the nanoclays did not lead to a change in the structure of the functional groups of PLA, but showed positive interaction between carbonyl groups of PLA and hydroxyl groups of modified C30B. Meanwhile, from XRD analysis it was observed that exfoliated structures can be obtained in bionanocomposites due to the plasticizing effect of scCO_2_ during the foaming process that even improves the dispersion of the nanoparticles and promotes exfoliation. This good dispersion producing the thermal stabilities of the nanocomposite foams was enhanced as compared to the neat PLA foams. The water uptake assay showed that the PLA- as well as PLA-C30B-loaded foams practically did not absorb water during the first week. Finally, all foams were completely disintegrated under composting conditions in less than one month and showed higher disintegration rates than the films of PLA and PLA-C30B bionanocomposites.

The results suggest that PLA bionanocomposite foams obtained by batch supercritical foaming represent a sustainable option to replace non-biodegradable expanded polystyrene and they are a promising option to be used in applications such as controlled release in food packaging.

## Figures and Tables

**Figure 1 polymers-14-04394-f001:**
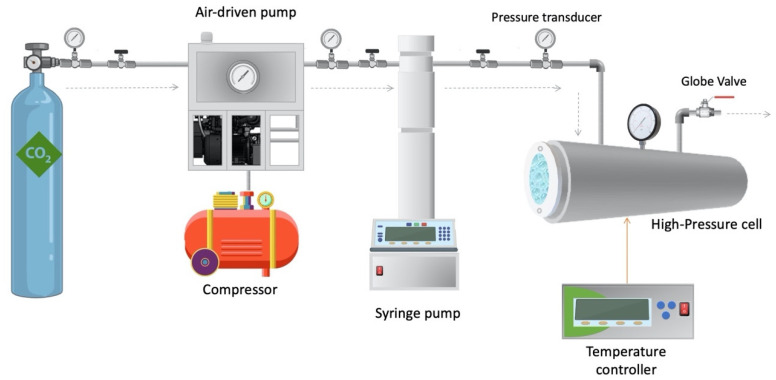
Outline of the experimental setup for supercritical batch foaming process.

**Figure 2 polymers-14-04394-f002:**
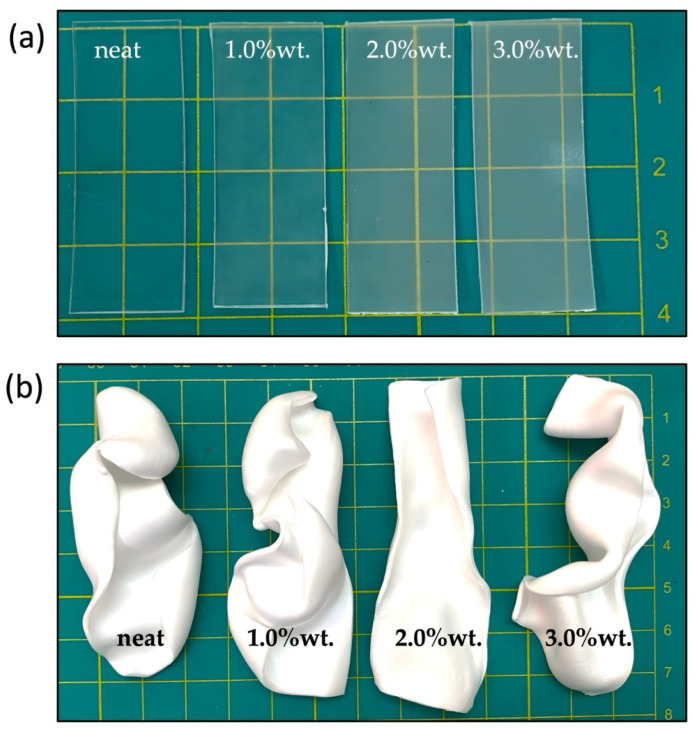
PLA and PLA/C30B bionanocomposite films (**a**) and foams (**b**).

**Figure 3 polymers-14-04394-f003:**
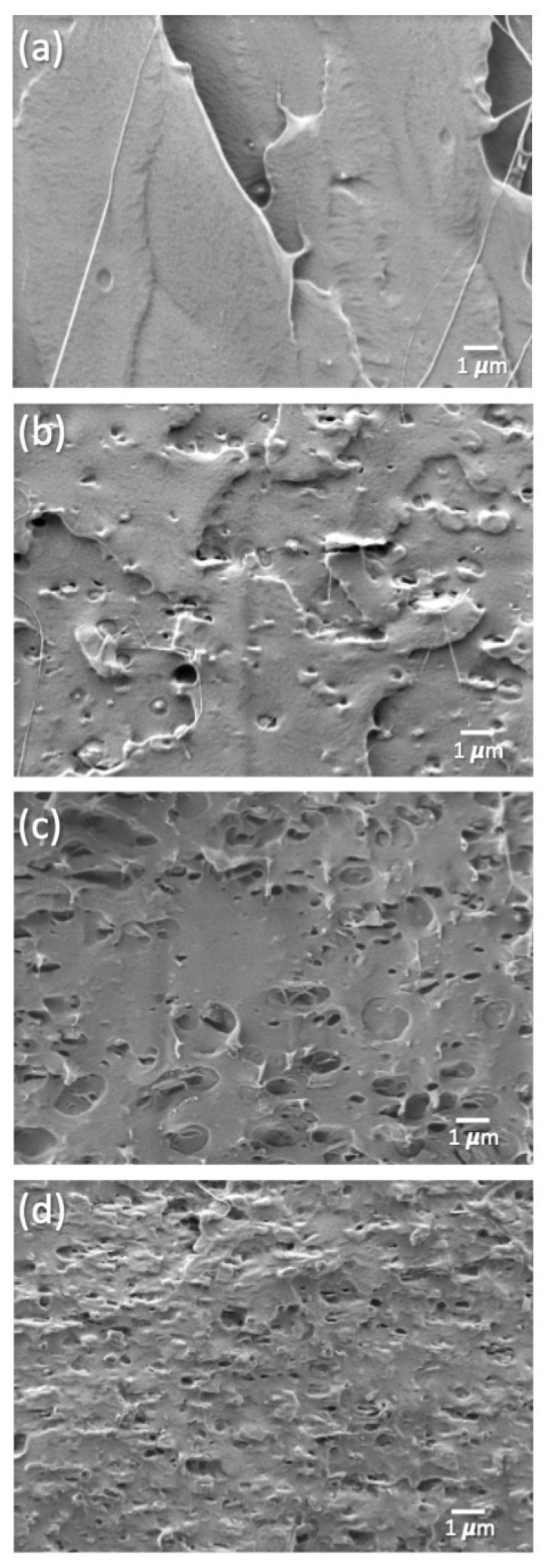
SEM micrographs of cross-section of freeze-fractured samples at 5.0 kx of PLA film samples. (**a**) PLA neat nanocomposites, (**b**) PLA 1% C30B, (**c**) PLA 2% C30B, and (**d**) PLA 3%C30B.

**Figure 4 polymers-14-04394-f004:**
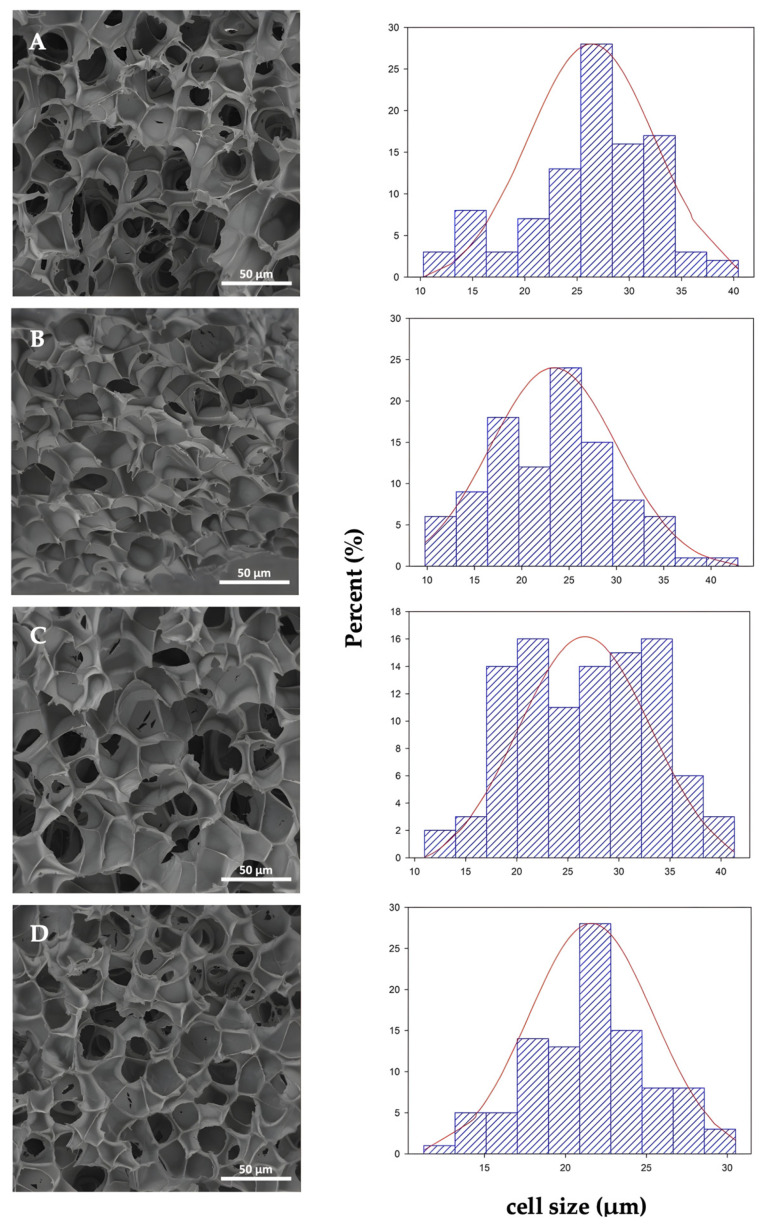
Left side: SEM micrographs of cross-sections of samples freeze-fractured at 1.0 kx of PLA foam samples (**A**) neat PLAf, (**B**) PLAf 1% C30B, (**C**) PLAf 2% C30B, and (**D**) PLAf 3% C30B nanocomposites, reprinted from Rojas et al. [43] under Creative Commons CC BY license.

**Figure 5 polymers-14-04394-f005:**
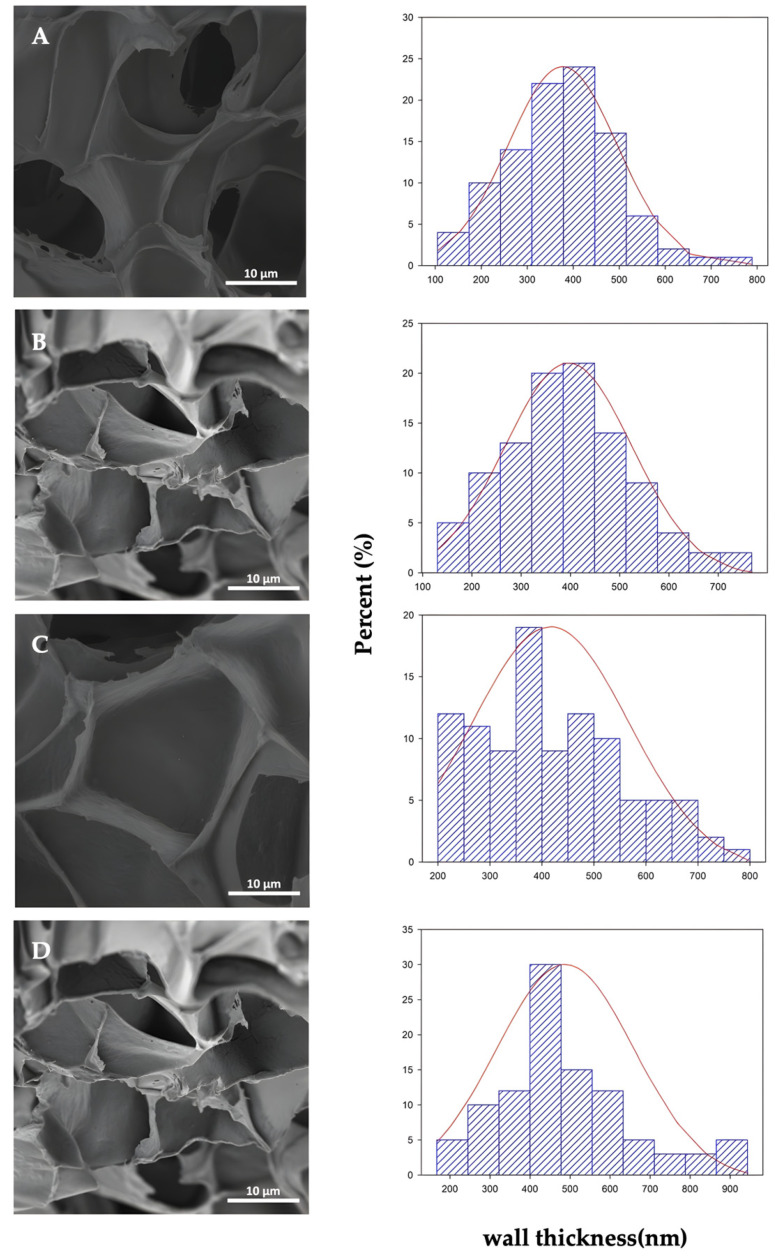
Left side: SEM micrographs of cross-sections of samples cryofractured at 4.0 kx and wall thickness distribution of PLA foam cells. PLA foam samples (**A**) neat PLAf, (**B**) PLAf 1% C30B, (**C**) PLAf 2% C30B, and (**D**) PLAf 3% C30B nanocomposites, reprinted from Rojas et al. [43] under Creative Commons CC BY license.

**Figure 6 polymers-14-04394-f006:**
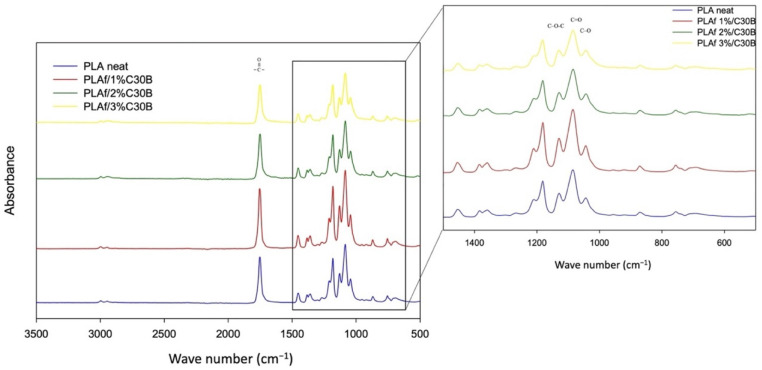
Fourier-transform Infrared Spectroscopy (FTIR) spectra of PLA bionanocomposite and neat foams.

**Figure 7 polymers-14-04394-f007:**
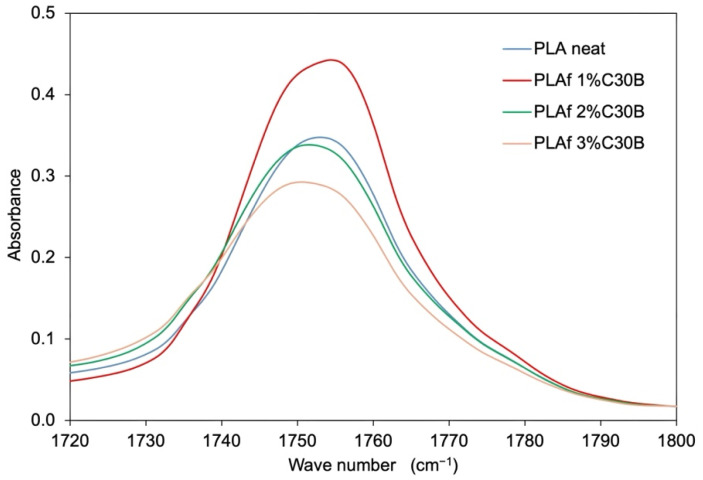
Fourier-transform Infrared Spectroscopy (FTIR) spectra between 1720 cm^−1^ and 1800 cm^−1^ of PLA bionanocomposite foams and neat PLA.

**Figure 8 polymers-14-04394-f008:**
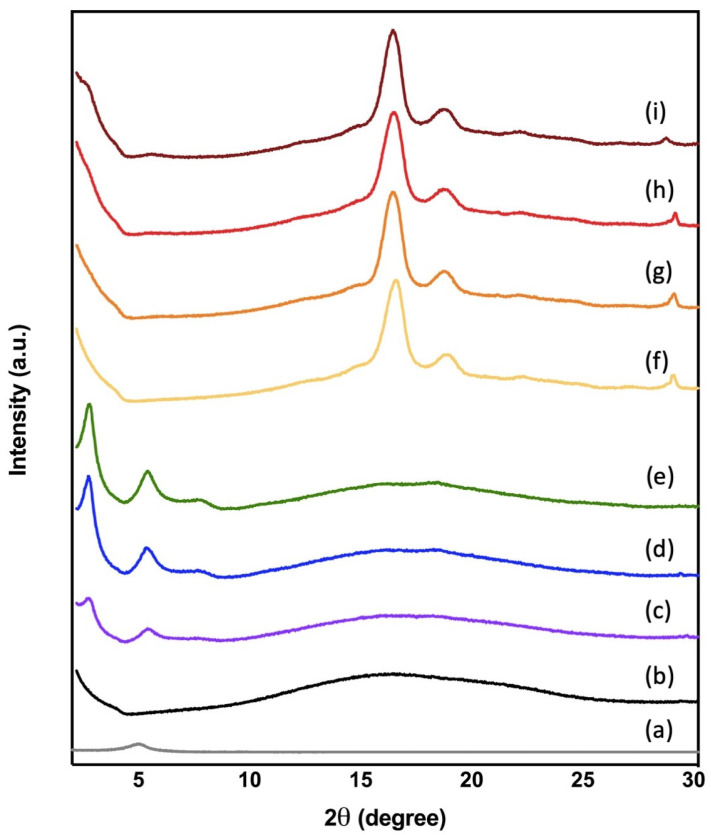
X-ray diffraction patterns of (a) Cloisite 30B, PLA film samples: (b) PLA neat, (c) PLA 1%C30B, (d) PLA 2%C30B, (e) PLA 3%C30B and PLA foam samples (f) PLAf neat, (g) PLAf 1%C30B, (h) PLAf 2%C30B, and (i) PLAf 3%C30B.

**Figure 9 polymers-14-04394-f009:**
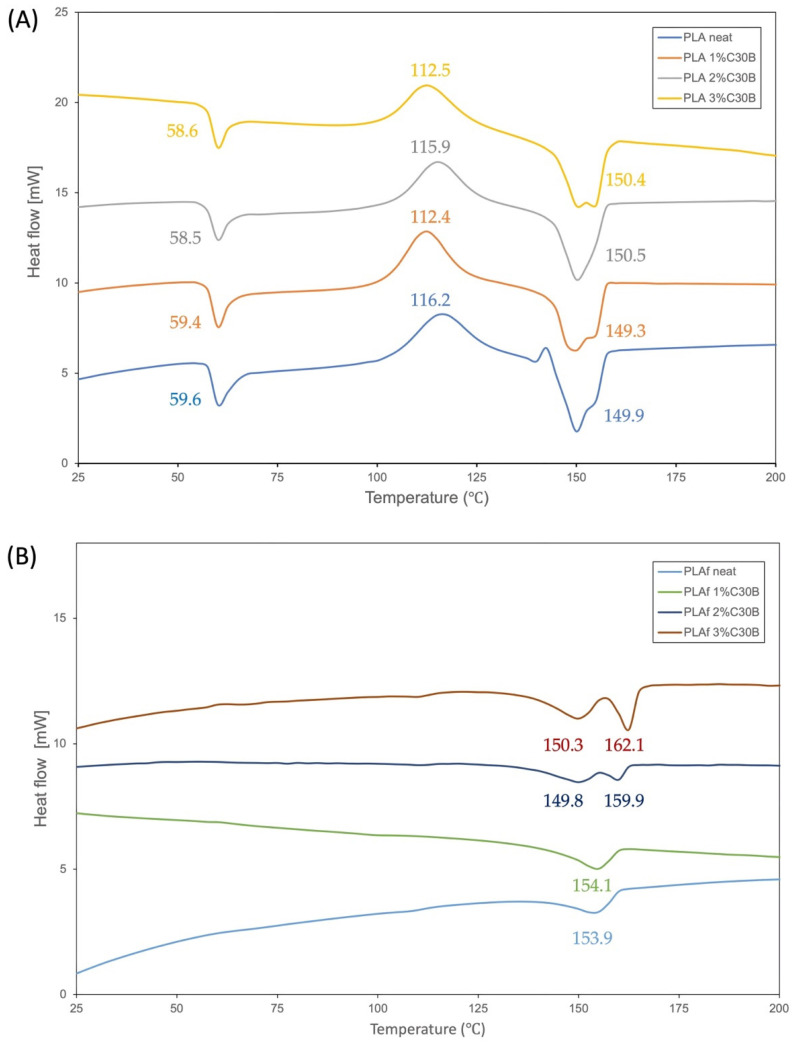
Differential Scanning Calorimetry thermograms of (**A**) PLA films and (**B**) PLA foams (PLAf) of neat and PLA bionanocomposites (Exo-up).

**Figure 10 polymers-14-04394-f010:**
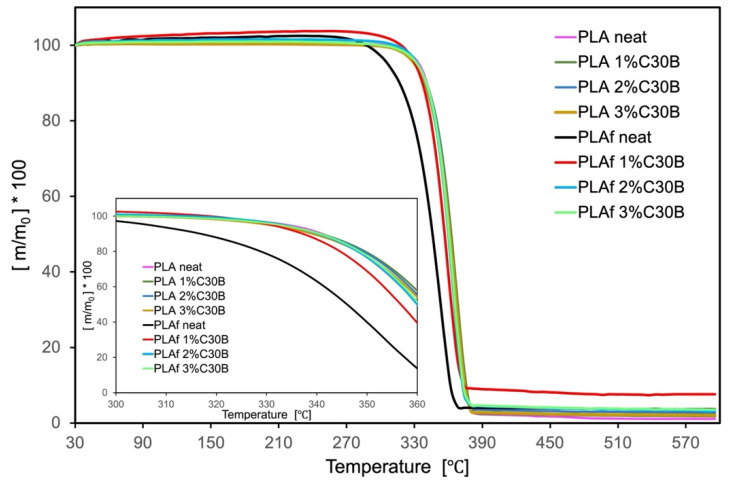
TGA and their derivative DTGA (insert) curves of PLA composites of films and foams.

**Figure 11 polymers-14-04394-f011:**
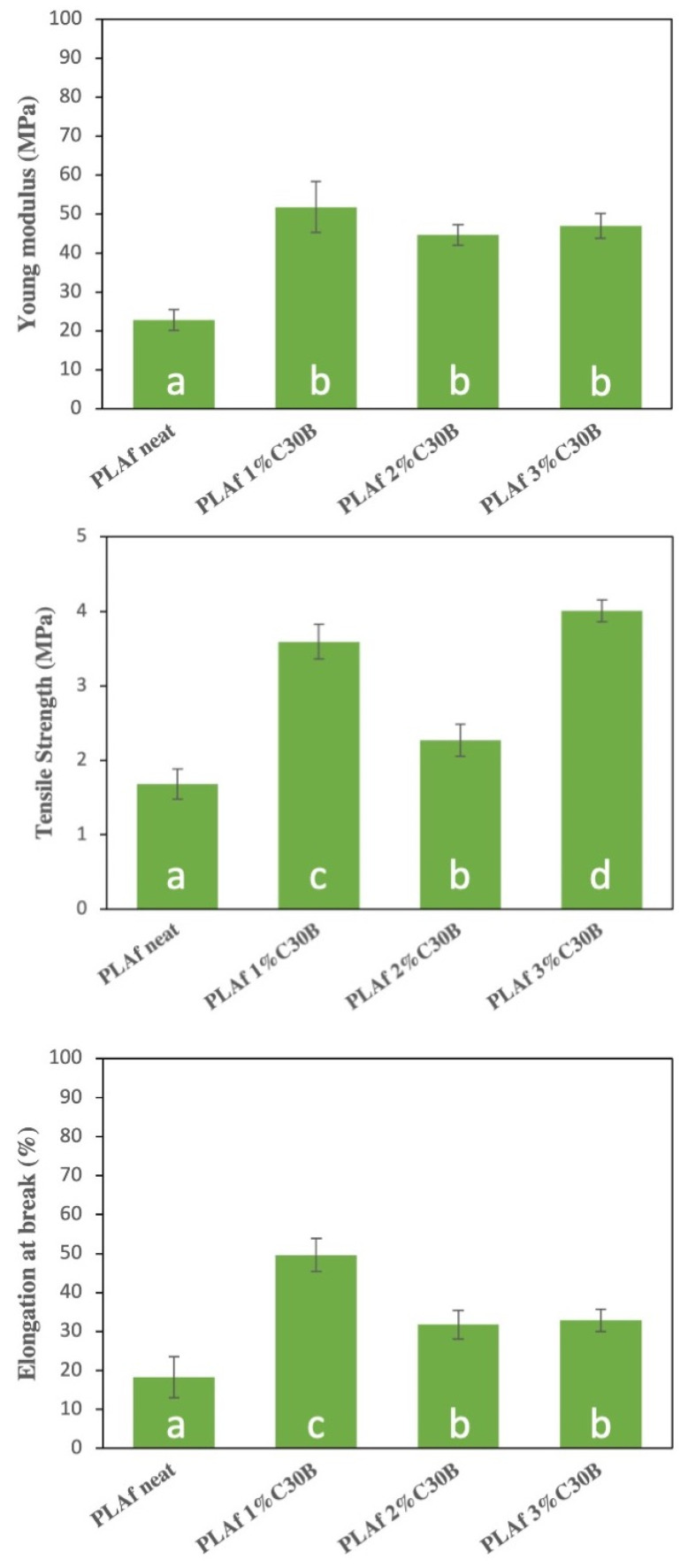
Mechanical properties of PLA and PLA bionanocomposite foams obtained from tensile test measurements. a, b Different superscripts within the same graph indicate significant differences between formulations (*p* < 0.05).

**Figure 12 polymers-14-04394-f012:**
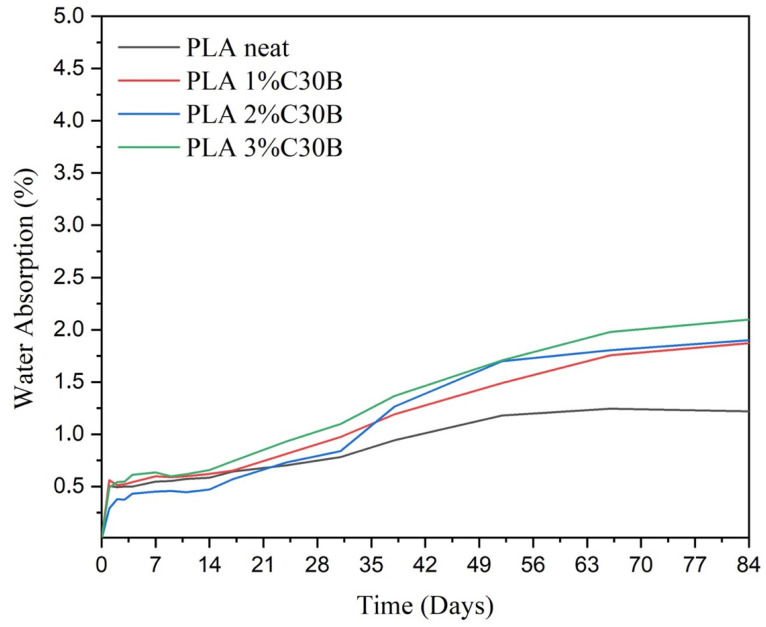
Water uptake of the PLA neat and PLA/C30B bionanocomposite films.

**Figure 13 polymers-14-04394-f013:**
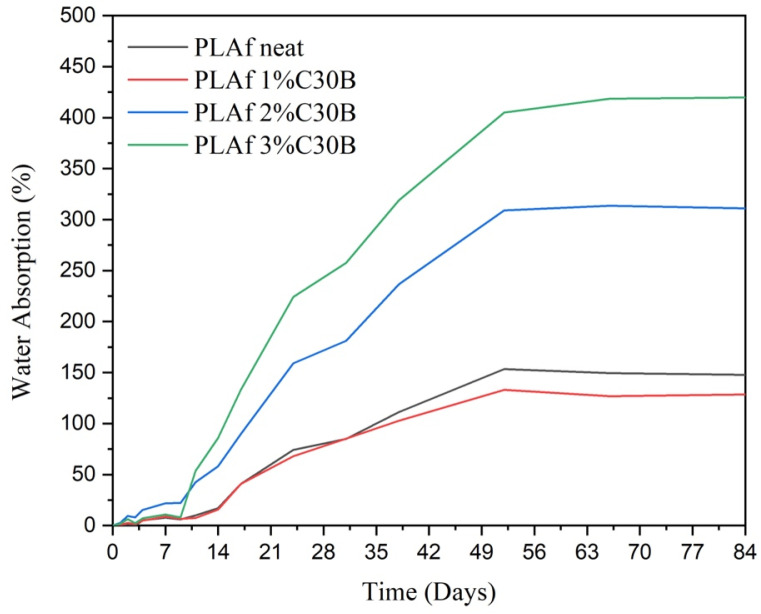
Water uptake of the PLA neat and PLA/C30B bionanocomposite foams.

**Figure 14 polymers-14-04394-f014:**
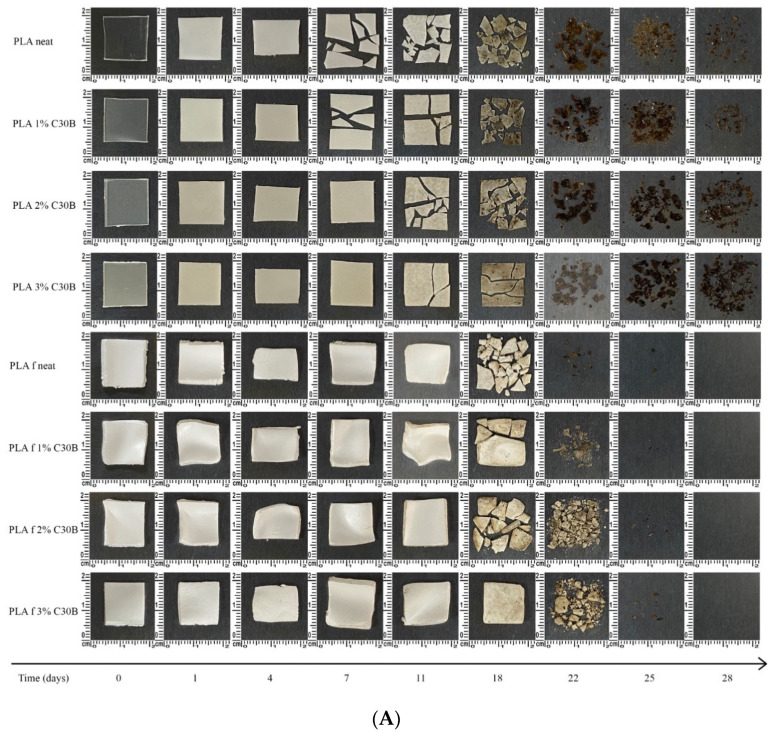
Disintegration under standard composting conditions of bionanocomposite film and foam samples. (**A**) visual appearance before and after different incubation days under composting conditions; (**B**) films’ and foams’ disintegration degree under composting conditions as a function of time.

**Table 1 polymers-14-04394-t001:** Mean values for intrinsic viscosity (η) and molecular weight (M_v_) of the PLA and PLA bionanocomposite films and foams (PLAf).

Sample	η (mL g^−1^)	$M_{v}$ (g mol^−1^)
*films*		
PLA neat	133.55	165,000
PLA 1% C30B	119.64	140,200
PLA 2% C30B	120.89	142,000
PLA 3% C30B	111.02	125,400
*foams*		
PLAf neat	121.72	144,750
PLAf 1% C30B	113.89	129,950
PLAf 2% C30B	105.82	117,750
PLAf 3% C30B	110.62	123,050

**Table 2 polymers-14-04394-t002:** Cell size (d), wall thickness (δ ), cell density (NC) and void fraction (V_f_).

Sample	d (μm)	δ (nm)	NC (× 10^11^ cell/cm^3^)	V_f_
PLAf neat	26.47 ± 6.11 ^a^	376 ± 119 ^a^	4.9	0.90
PLAf 1% C30B	23.35 ± 6,71 ^b^	395 ± 129 ^a^	7.1	0.91
PLAf 2% C30B	26.69 ± 6.52 ^a^	421 ± 136 ^a^	4.7	0.89
PLAf 3% C30B	21.58 ± 3.84 ^b^	486 ± 172 ^b^	8.2	0.83

a, b Different superscripts within the same column indicate significant differences between formulations (*p* < 0.05).

**Table 3 polymers-14-04394-t003:** TGA parameters of PLA bionanocomposite films and foams.

Sample	Td, 5% (°C)	T_max_ (°C)
*films*		
PLA neat	333.9	363.9
PLA1% C30B	332.1	366.4
PLA2% C30B	331.7	365.1
PLA3% C30B	330.0	365.3
*foams*		
PLAf neat	306.4	356.7
PLAf 1% C30B	330.5	364.2
PLAf 2% C30B	332.9	366.1
PLAf 3% C30B	332.3	365.3

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
