# Peer review of "Processing Compostable PLA/Organoclay Bionanocomposite Foams by Supercritical CO2 Foaming for Sustainable Food Packaging"

_polymers, 2022, doi:10.3390/polym14204394_

Round 1

Reviewer 1 Report

Dear authors,

I consider your paper entitled "Processing Compostable PLA/Organoclay Bionanocomposite Foams by CO2 Supercritical Foaming for Sustainable Food Packaging" is appropriate for being published in Polymers.

Author Response

We thank the reviewer for considering our manuscript suitable for its publication in its current form.

Reviewer 2 Report

Processing Compostable PLA/Organoclay Bionanocomposite 2 Foams by CO2 Supercritical Foaming for Sustainable Food 3 Packaging 4

By Simón Faba, Marina P. Arrieta, Ángel Agüero, Alejandra Torres, Julio Romero, Adrián Rojas and María José Galotto

The manuscript is aimed to obtain compostable bionanocomposite foams based on PLA and organoclay (C30B) by foaming using supercritical carbon dioxide.

In my opinion, the manuscript is not suitable for publication in “Polymers”. Below you find my arguments:

1. The method used is not new (as the authors said): the preparation method for PLA films and composites was previously published by the same group of authors in [Rojas, A.; Torres, A.; López de Dicastillo, C.; Velásquez, E.; Villegas, C.; Faba, S.; Rivera, P.; Guarda, A.; Romero, J.; Galotto, M.J. Foaming with scCO2 and Impregnation with Cinnamaldehyde of PLA Nanocomposites for Food Packaging. Processes 2022, 10, 376], but no references were given in the present manuscript.

2. Although the conditions for PLA foaming are presented slightly changed (130oC instead of 135 oC, an 30 minutes instead of 35 minutes), the SEM images and FTIR results presented in the manuscript are the same as the ones presented in their previous paper (how is that possible?).

3. The experimental setup presented in Figure 1 is already published twice, with some small changes (in Rojas, A.; Torres, A.; López de Dicastillo, C.; Velásquez, E.; Villegas, C.; Faba, S.; Rivera, P.; Guarda, A.; Romero, J.; Galotto, M.J. Foaming with scCO2 and Impregnation with Cinnamaldehyde of PLA Nanocomposites for Food Packaging. Processes 2022, 10, 376; and C.Villegas, M.P.Arrieta, A.Rojas, A.Torres, S.Faba, M.J.Toledo, M.A.Gutierrez, E.Zavalla, J.Romero, M.J.Galotto, X.Valenzuela, PLA/organoclay bionanocomposites impregnated with thymol and cinnamaldehyde by supercritical impregnation for active and sustainable food packaging, Composites Part B, 176 (2019) 107336].

4. The manuscript is not well organized and the data are presented chaotically:

4.1. Figure 2: the images are not defined

4.2. Table 1: the authors didn’t explained why PLA/2C30B had a different behavior

4.3. SEM results: the authors gave 3 sets of images, at different magnitudes, for the same samples, 2 of them already being published already in their previous paper [Rojas, A.; Torres, A.; López de Dicastillo, C.; Velásquez, E.; Villegas, C.; Faba, S.; Rivera, P.; Guarda, A.; Romero, J.; Galotto, M.J. Foaming with scCO2 and Impregnation with Cinnamaldehyde of PLA Nanocomposites for Food Packaging. Processes 2022, 10, 376].

4.4. FTIR results:

- the authors drew some conclusions that cannot be observed in the spectra, like: R335 “The carbonyl group, centered at 1753 cm-1 in neat PLA sample, was shifted to somewhat lower wavenumbers in bionanocomposites (1748 cm-1) probably due to the crystalline carbonyl stretching as a consequence of the C30B nucleating effect.”. No band shift can be observed from the FTIR spectra

- the authors made correlation between bands intensity, but didn’t specify if the spectra are normalized, otherwise a comparison between band intensities is not correct.

4.5. XRD result: no crystallographic planes assigned

4.6. DSC results:

- no endo/exo direction is indicated on the “y” axe

- the discussion films/foams is mixed up: e.g. the authors said R406: “Regarding the foams, only in the PLA 3%C30B samples it was observed a significant increase of Tg value and slight decrease of Tcc and Tm, respect to the neat PLA foam”, when from DSC figures of foam samples no glass transition temperature was observed.

4.7. TGA results: DTG images are not clear, and hard to follow

4.8. Mechanical properties: clays are well known to act as reinforcing agents, so is no explanation for increasing the elongation at break related to enhanced Young Modulus and tensile strength.

5. The manuscript needs substantial revision in terms of English language, grammatical corrections (the entire manuscript needs to be proofread by a native speaker)

 Taking into consideration all the above, I do not recommend the manuscript for publication in “Polymers”.

Author Response

Find enclosed the response to reviewer 2

Reviewer 3 Report

In this contribution, the authors prepared bionanocomposite foams based on poly(lactic acid)  and organoclay using a supercritical COfoaming method. Those bionanocomposite foams  show good mechanical performance, controllable water uptake, and compostability, and thus  are promising for applications, such as food packing. The thorough characterizations and  convincing results support the authors’ conclusions. Furthermore, this research is inspiring for  the readership of Polymers. Therefore, I would recommend its publishing if the following  questions and comments are addressed.  1. In Line 184, the onset decomposition temperature (Tonset) is determined at 5% of mass loss.  However, ASTM E2550 defines Tonset as the point on the TGA curve where a deflection is first  observed from the established baseline prior to the thermal event. The temperature at 5% of  mass loss is usually denoted as Td,5% 2. Besides the macropores with sizes above 20 micrometers, does the supercritical CO foaming and crystallization of PLA create micropores and mesopores on the wall of cells, as  exemplified in the works in DOI: 10.1002/adma.200300380? This is also to be verified using  high-resolution SEM or BET.  3. In Figure 2a and b, the notations do not match with samples. Supposedly, those films and  foams are made of neat PLA, PLA with 1%, 2%, and 3% of C30B.  4. The details of how to estimate densities using the method of buoyancy need elaboration  (Line 159). Does this method employ the buoyancy when immersing foams in a liquid, e.g.,  water? Does this liquid intrude into cells and affect the estimation of densities and porosity?  5. In Figure 8, all foams (f, g, h, and i) show a peak near 2θ of 29°. What are those peaks  attributed to? In another of your contribution, those peaks are not observed in compositions  consisting of PLA and organoclay. (Compos. Part B Eng2019176, 107336,  doi:10.1016/j.compositesb.2019.107336)  6. When measuring the water uptake of bionanocomposite foams, is water absorbed into the  wall or cells? If water is absorbed into the wall, why the water uptake of foams is profoundly  higher than that of saturated films? If water is absorbed into cells, why does neat PLA foam  with the largest porosity show the lowest water uptake? In addition, the water uptake of neat  PLA film is not 1.2% after 3 weeks (Line 457).  7. Given the foaming process requires scCOsaturation and COdiffusion in the polymer  matrix (Line 290 and 259), does the uniformity of cell size and wall thickness depend on the  thickness/dimension of foams on an industrial scale? For example, larger pores appear on the  surface of monolith foam due to faster scCOsaturation and COdiffusion, but smaller pores  appear inside the monolith foam.

Author Response

find enclosed the response to reviewer 3

Round 2

Reviewer 2 Report

Unfortunately, I cannot recommend the publication of the manuscript in “Polymers”. Below you find my arguments:

1. The authors tried to induce the idea that the samples were synthesized in different conditions (time and temperature), when in fact results on these samples were published before, in [Rojas, A.; Torres, A.; López de Dicastillo, C.; Velásquez, E.; Villegas, C.; Faba, S.; Rivera, P.; Guarda, A.; Romero, J.; Galotto, M.J. Foaming with scCO2 and Impregnation with Cinnamaldehyde of PLA Nanocomposites for Food Packaging. Processes 2022, 10, 376].

2. I think that the authors should have only referred to the article in which the SEM and FTIR data are already published (not try to publish them again, in a different way), and in the present manuscript only the new elements should be presented. This is not a review in order to cite previously published data, even under Creative Commons license.

3. The same regarding the experimental setup; it should only cite the articles in which it was previously presented, namely Rojas, A.; Torres, A.; López de Dicastillo, C.; Velásquez, E.; Villegas, C.; Faba, S.; Rivera, P.; Guarda, A.; Romero, J.; Galotto, M.J. Foaming with scCO2 and Impregnation with Cinnamaldehyde of PLA Nanocomposites for Food Packaging. Processes 2022, 10, 376; and C.Villegas, M.P.Arrieta, A.Rojas, A.Torres, S.Faba, M.J.Toledo, M.A.Gutierrez, E.Zavalla, J.Romero, M.J.Galotto, X.Valenzuela, PLA/organoclay bionanocomposites impregnated with thymol and cinnamaldehyde by supercritical impregnation for active and sustainable food packaging, Composites Part B, 176 (2019) 107336.

4. The authors changed Figure 10; however it seems to be a mistake. Upon careful examination of the graph inside, which should represent the enlarged image of the 300-360 oC range, it does not correspond to the image of the initial graph. Especially observe the curves for PLA/neat (black) and PLA 1% C30B (red).

5. Also the authors are not constant in their affirmation. They changed their opinion based on the same results. First they said “Regarding the foams, only in the PLA 3%C30B samples it was observed a significant increase of Tg value and slight decrease of Tcc and Tm, respect to the neat PLA foam” and after revision “Regarding the foams, in all samples the cold crystallization temperature did not take place, suggesting that the material completely crystalized during supercritical CO2 process”, but the data are the same. Nevertheless, “temperature” cannot “take place”. The authors probably meant to “cold crystallization process”.

I recommend the reorganization of the manuscript (without SEM, FTIR and experimental setup) and resubmission as a different manuscript. In this form I cannot recommend the publication in “Polymers”.